# Buy-in-Bulk Active Learning

**Liu Yang**
Machine Learning Department,
Carnegie Mellon University
liuy@cs.cmu.edu

**Jaime Carbonell**
Language Technologies Institute,
Carnegie Mellon University
jgc@cs.cmu.edu

## Abstract

In many practical applications of active learning, it is more cost-effective to request labels in large batches, rather than one-at-a-time. This is because the cost of labeling a large batch of examples at once is often sublinear in the number of examples in the batch. In this work, we study the label complexity of active learning algorithms that request labels in a given number of batches, as well as the tradeoff between the total number of queries and the number of rounds allowed. We additionally study the total cost sufficient for learning, for an abstract notion of the cost of requesting the labels of a given number of examples at once. In particular, we find that for sublinear cost functions, it is often desirable to request labels in large batches (i.e., buying in bulk); although this may increase the total number of labels requested, it reduces the total cost required for learning.

## 1 Introduction

In many practical applications of active learning, the cost to acquire a large batch of labels at once is significantly less than the cost of the same number of sequential rounds of individual label requests. This is true for both practical reasons (overhead time for start-up, reserving equipment in discrete time-blocks, multiple labelers working in parallel, etc.) and for computational reasons (e.g., time to update the learner's hypothesis and select the next examples may be large). Consider making one vs multiple hematological diagnostic tests on an out-patient. There are fixed up-front costs: bringing the patient in for testing, drawing and storing the blood, entring the information in the hospital record system, etc. And there are variable costs, per specific test. Consider a microarray assay for gene expression data. There is a fixed cost in setting up and running the microarray, but virtually no incremental cost as to the number of samples, just a constraint on the max allowed. Either of the above conditions are often the case in scientific experiments (e.g., [1]), As a different example, consider calling a focused group of experts to address questions w.r.t new product design or introduction. There is a fixed cost in forming the group (determine membership, contract, travel, etc.), and a incremental per-question cost. The common abstraction in such real-world versions of "oracles" is that learning can buy-in-bulk to advantage because oracles charge either per batch (answering a batch of questions for the same cost as answering a single question up to a batch maximum), or the cost per batch is $ax^p + b$, where $b$ is the set-up cost, $x$ is the number of queries, and $p = 1$ or $p < 1$ (for the case where practice yields efficiency).

Often we have other tradeoffs, such as delay vs testing cost. For instance in a medical diagnosis case, the most cost-effective way to minimize diagnostic tests is purely sequential active learning, where each test may rule out a set of hypotheses (diagnoses) and informs the next test to perform. But a patient suffering from a serious disease may worsen while sequential tests are being conducted. Hence batch testing makes sense if the batch can be tested in parallel. In general one can convert delay into a second cost factor and optimize for batch size that minimizes a combination of total delay and the sum of the costs for the individual tests. Parallelizing means more tests would be needed, since we lack the benefit of earlier tests to rule out future ones. In order to perform this

batch-size optimization we also need to estimate the number of redundant tests incurred by turning a sequence into a shorter sequence of batches.

For the reasons cited above, it can be very useful in practice to generalize active learning to active-batch learning, with buy-in-bulk discounts. This paper developes a theoretical framework exploring the bounds and sample compelxity of active buy-in-bulk machine learning, and analyzing the trade-off that can be achieved between the number of batches and the total number of queries required for accurate learning.

In another example, if we have many labelers (virtually unlimited) operating in parallel, but must pay for each query, and the amount of time to get back the answer to each query is considered independent with some distribution, it may often be the case that the expected amount of time needed to get back the answers to $m$ queries is sublinear in $m$, so that if the "cost" is a function of both the payment amounts and the time, it might sometimes be less costly to submit multiple queries to be labeled in parallel. In scenarios such as those mentioned above, a batch mode active learning strategy is desirable, rather than a method that selects instances to be labeled one-at-a-time.

There have recently been several attempts to construct heuristic approaches to the batch mode active learning problem (e.g., [2]). However, theoretical analysis has been largely lacking. In contrast, there has recently been significant progress in understanding the advantages of fully-sequential active learning (e.g., [3, 4, 5, 6, 7]). In the present work, we are interested in extending the techniques used for the fully-sequential active learning model, studying natural analogues of them for the batch-model active learning model.

Formally, we are interested in two quantities: the sample complexity and the total cost. The sample complexity refers to the number of label requests used by the algorithm. We expect batch-mode active learning methods to use *more* label requests than their fully-sequential cousins. On the other hand, if the *cost* to obtain a batch of labels is *sublinear* in the size of the batch, then we may sometimes expect the total cost used by a batch-mode learning method to be significantly *less* than the analogous fully-sequential algorithms, which request labels individually.

## 2 Definitions and Notation

As in the usual statistical learning problem, there is a standard Borel space $\mathcal{X}$, called the instance space, and a set $\mathbb{C}$ of measurable classifiers $h : \mathcal{X} \to \{-1, +1\}$, called the concept space. Throughout, we suppose that the VC dimension of $\mathbb{C}$, denoted $d$ below, is finite.

In the learning problem, there is an unobservable distribution $\mathcal{D}_{XY}$ over $\mathcal{X} \times \{-1, +1\}$. Based on this quantity, we let $\mathcal{Z} = \{(X_t, Y_t)\}_{t=1}^{\infty}$ denote an infinite sequence of independent $\mathcal{D}_{XY}$-distributed random variables. We also denote by $\mathcal{Z}_t = \{(X_1, Y_1), (X_2, Y_2), \dots, (X_t, Y_t)\}$ the first $t$ such labeled examples. Additionally denote by $\mathcal{D}_X$ the marginal distribution of $\mathcal{D}_{XY}$ over $\mathcal{X}$. For a classifier $h : \mathcal{X} \to \{-1, +1\}$, denote $\text{er}(h) = P_{(X,Y) \sim \mathcal{D}_{XY}}(h(X) \neq Y)$, the *error rate* of $h$. Additionally, for $m \in \mathbb{N}$ and $Q \in (\mathcal{X} \times \{-1, +1\})^m$, let $\text{er}(h; Q) = \frac{1}{|Q|} \sum_{(x,y) \in Q} \mathbb{I}[h(x) \neq y]$, the *empirical error rate* of $h$. In the special case that $Q = \mathcal{Z}_m$, abbreviate $\text{er}_m(h) = \text{er}(h; Q)$. For $r > 0$, define $\text{B}(h, r) = \{g \in \mathbb{C} : \mathcal{D}_X(x : h(x) \neq g(x)) \leq r\}$. For any $\mathcal{H} \subseteq \mathbb{C}$, define $\text{DIS}(\mathcal{H}) = \{x \in \mathcal{X} : \exists h, g \in \mathcal{H} \text{ s.t. } h(x) \neq g(x)\}$. We also denote by $\eta(x) = P(Y = +1 | X = x)$, where $(X, Y) \sim \mathcal{D}_{XY}$, and let $h^*(x) = \text{sign}(\eta(x) - 1/2)$ denote the *Bayes optimal classifier*.

In the active learning protocol, the algorithm has direct access to the $X_t$ sequence, but must request to observe each label $Y_t$, sequentially. The algorithm asks up to a specified number of label requests $n$ (the *budget*), and then halts and returns a classifier. We are particularly interested in determining, for a given algorithm, how large this number of label requests needs to be in order to guarantee small error rate with high probability, a value known as the *label complexity*. In the present work, we are also interested in the *cost* expended by the algorithm. Specifically, in this context, there is a cost function $c : \mathbb{N} \to (0, \infty)$, and to request the labels $\{Y_{i_1}, Y_{i_2}, \dots, Y_{i_m}\}$ of $m$ examples $\{X_{i_1}, X_{i_2}, \dots, X_{i_m}\}$ at once requires the algorithm to pay $c(m)$; we are then interested in the sum of these costs, over all *batches* of label requests made by the algorithm. Depending on the form of the cost function, minimizing the cost of learning may actually require the algorithm to request labels in batches, which we expect would actually increase the total number of label requests.

To help quantify the label complexity and cost complexity, we make use of the following definition, due to [6, 7].

**Definition 2.1.** *[6, 7] Define the disagreement coefficient of $h^*$ as*

$$\theta(\epsilon) = \sup_{r > \epsilon} \frac{\mathcal{D}_X\left(\mathrm{DIS}(\mathrm{B}(h^*, r))\right)}{r}.$$

## 3 Buy-in-Bulk Active Learning in the Realizable Case: k-batch CAL

We begin our anlaysis with the simplest case: namely, the realizable case, with a fixed prespecified number of batches. We are then interested in quantifying the label complexity for such a scenario.

Formally, in this section we suppose $h^* \in \mathbb{C}$ and $\mathrm{er}(h^*) = 0$. This is refered to as the *realizable case*. We first review a well-known method for active learning in the realizable case, refered to as CAL after its discoverers Cohn, Atlas, and Ladner [8].

---

Algorithm: CAL($n$)
1. $t \leftarrow 0, m \leftarrow 0, \mathcal{Q} \leftarrow \emptyset$
2. While $t < n$
3.    $m \leftarrow m + 1$
4.    If $\max\limits_{y \in \{-1, +1\}} \min\limits_{h \in \mathbb{C}} \mathrm{er}(h; \mathcal{Q} \cup \{(X_m, y)\}) = 0$
5.       Request $Y_m$, let $\mathcal{Q} \leftarrow \mathcal{Q} \cup \{(X_m, Y_m)\}, t \leftarrow t + 1$
6. Return $\hat{h} = \mathrm{argmin}_{h \in \mathbb{C}} \mathrm{er}(h; \mathcal{Q})$

---

The label complexity of CAL is known to be $O\left(\theta(\epsilon)(d \log(\theta(\epsilon)) + \log(\log(1/\epsilon)/\delta)) \log(1/\epsilon)\right)$ [7]. That is, some $n$ of this size suffices to guarantee that, with probability $1 - \delta$, the returned classifier $\hat{h}$ has $\mathrm{er}(\hat{h}) \leq \epsilon$.

One particularly simple way to modify this algorithm to make it batch-based is to simply divide up the budget into equal batch sizes. This yields the following method, which we refer to as $k$-batch CAL, where $k \in \{1, \ldots, n\}$.

---

Algorithm: $k$-batch CAL($n$)
1. Let $Q \leftarrow \{\}, b \leftarrow 2, V \leftarrow \mathbb{C}$
2. For $m = 1, 2, \ldots$
3.    If $X_m \in DIS(V)$
4.       $Q \leftarrow Q \cup \{X_m\}$
5.    If $|Q| = \lfloor n/k \rfloor$
6.       Request the labels of examples in $Q$
7.       Let $L$ be the corresponding labeled examples
8.       $V \leftarrow \{h \in V : \mathrm{er}(h; L) = 0\}$
9.       $b \leftarrow b + 1$ and $Q \leftarrow \emptyset$
10.     If $b > k$, Return any $\hat{h} \in V$

---

We expect the label complexity of $k$-batch CAL to somehow interpolate between passive learning (at $k = 1$) and the label complexity of CAL (at $k = n$). Indeed, the following theorem bounds the label complexity of $k$-batch CAL by a function that exhibits this interpolation behavior with respect to the known upper bounds for these two cases.

**Theorem 3.1.** *In the realizable case, for some*

$$\lambda(\epsilon, \delta) = O\left(k\epsilon^{-1/k}\theta(\epsilon)^{1-1/k}(d \log(1/\epsilon) + \log(1/\delta))\right),$$

*for any $n \geq \lambda(\epsilon, \delta)$, with probability at least $1 - \delta$, running $k$-batch CAL with budget $n$ produces a classifier $\hat{h}$ with $\mathrm{er}(\hat{h}) \leq \epsilon$.*

*Proof.* Let $M = \lfloor n/k \rfloor$. Define $V_0 = \mathbb{C}$ and $i_{0M} = 0$. Generally, for $b \geq 1$, let $i_{b1}, i_{b2}, \ldots, i_{bM}$ denote the indices $i$ of the first $M$ points $X_i \in \mathrm{DIS}(V_{b-1})$ for which $i > i_{(b-1)M}$, and let $V_b = \{h \in$

$V_{b-1} : \forall j \leq M, h(X_{i_{b_j}}) = h^*(X_{i_{b_j}})\}$. These correspond to the version space at the conclusion of batch $b$ in the $k$-batch CAL algorithm.

Note that $X_{i_{b_1}}, \ldots, X_{i_{b_M}}$ are conditionally iid given $V_{b-1}$, with distribution of $X$ given $X \in DIS(V_{b-1})$. Thus, the PAC bound of [9] implies that, for some constant $c \in (0, \infty)$, with probability $\geq 1 - \delta/k$,

$$V_b \subseteq B\left(h^*, c\frac{d\log(M/d) + \log(k/\delta)}{M} P(DIS(V_{b-1}))\right).$$

By a union bound, the above holds for all $b \leq k$ with probability $\geq 1 - \delta$; suppose this is the case. Since $P(DIS(V_{b-1})) \leq \theta(\epsilon)\max\{\epsilon, \max_{h \in V_{b-1}} er(h)\}$, and any $b$ with $\max_{h \in V_{b-1}} er(h) \leq \epsilon$ would also have $\max_{h \in V_b} er(h) \leq \epsilon$, we have

$$\max_{h \in V_b} er(h) \leq \max\left\{\epsilon, c\frac{d\log(M/d) + \log(k/\delta)}{M} \theta(\epsilon) \max_{h \in V_{b-1}} er(h))\right\}.$$

Noting that $P(DIS(V_0)) \leq 1$ implies $V_1 \subseteq B\left(h^*, c\frac{d\log(M/d)+\log(k/\delta)}{M}\right)$, by induction we have

$$\max_{h \in V_k} er(h) \leq \max\left\{\epsilon, \left(c\frac{d\log(M/d) + \log(k/\delta)}{M}\right)^k \theta(\epsilon)^{k-1}\right\}.$$

For some constant $c' > 0$, any $M \geq c'\frac{\theta(\epsilon)^{\frac{k-1}{k}}}{\epsilon^{1/k}}\left(d\log\frac{1}{\epsilon} + \log(k/\delta)\right)$ makes the right hand side $\leq \epsilon$. Since $M = \lfloor n/k \rfloor$, it suffices to have $n \geq k\left(1 + c'\frac{\theta(\epsilon)^{\frac{k-1}{k}}}{\epsilon^{1/k}}\left(d\log\frac{1}{\epsilon} + \log(k/\delta)\right)\right)$. $\qquad\square$

Theorem 3.1 has the property that, when the disagreement coefficient is small, the stated bound on the total number of label requests sufficient for learning is a decreasing function of $k$. This makes sense, since $\theta(\epsilon)$ small would imply that fully-sequential active learning is much better than passive learning. Small values of $k$ correspond to more passive-like behavior, while larger values of $k$ take fuller advantage of the sequential nature of active learning. In particular, when $k = 1$, we recover a well-known label complexity bound for passive learning by empirical risk minimization [10]. In contrast, when $k = \log(1/\epsilon)$, the $\epsilon^{-1/k}$ factor is $e$ (constant), and the rest of the bound is at most $O(\theta(\epsilon)(d\log(1/\epsilon) + \log(1/\delta))\log(1/\epsilon))$, which is (up to a $\log$ factor) a well-known bound on the label complexity of CAL for active learning [7] (a slight refinement of the proof would in fact recover the exact bound of [7] for this case); for $k$ larger than $\log(1/\epsilon)$, the label complexity can only improve; for instance, consider that upon reaching a given data point $X_m$ in the data stream, if $V$ is the version space in $k$-batch CAL (for some $k$), and $V'$ is the version space in $2k$-batch CAL, then we have $V' \subseteq V$ (supposing $n$ is a multiple of $2k$), so that $X_m \in DIS(V')$ only if $X_m \in DIS(V)$. Note that even $k = 2$ can sometimes provide significant reductions in label complexity over passive learning: for instance, by a factor proportional to $1/\sqrt{\epsilon}$ in the case that $\theta(\epsilon)$ is bounded by a finite constant.

## 4 Batch Mode Active Learning with Tsybakov noise

The above analysis was for the realizable case. While this provides a particularly clean and simple analysis, it is not sufficiently broad to cover many realistic learning applications. To move beyond the realizable case, we need to allow the labels to be noisy, so that $er(h^*) > 0$. One popular noise model in the statistical learning theory literature is Tsybakov noise, which is defined as follows.

**Definition 4.1.** *[11] The distribution $\mathcal{D}_{XY}$ satisfies* Tsybakov noise *if $h^* \in \mathbb{C}$, and for some $c > 0$ and $\alpha \in [0, 1]$,*

$$\forall t > 0, \mathbb{P}(|\eta(x) - 1/2| < t) < c_1 t^{\frac{\alpha}{1-\alpha}},$$

*equivalently, $\forall h, P(h(x) \neq h^*(x)) \leq c_2(er(h) - er(h^*))^\alpha$, where $c_1$ and $c_2$ are constants.*

Supposing $\mathcal{D}_{XY}$ satisfies Tsybakov noise, we define a quantity

$$\mathcal{E}_m = c_3\left(\frac{d\log(m/d) + \log(km/\delta)}{m}\right)^{\frac{1}{2-\alpha}}.$$

based on a standard generalization bound for passive learning [12]. Specifically, [12] have shown that, for any $V \subseteq \mathbb{C}$, with probability at least $1 - \delta/(4km^2)$,

$$\sup_{h,g \in V} |(\mathrm{er}(h) - \mathrm{er}(g)) - (\mathrm{er}_m(h) - \mathrm{er}_m(g))| < \mathcal{E}_m. \tag{1}$$

Consider the following modification of $k$-batch CAL, designed to be robust to Tsybakov noise. We refer to this method as $k$-batch Robust CAL, where $k \in \{1, \ldots, n\}$.

---

Algorithm: $k$-batch Robust CAL$(n)$
1. Let $Q \leftarrow \{\}$, $b \leftarrow 1$, $V \leftarrow \mathbb{C}$, $m_1 \leftarrow 0$
2. For $m = 1, 2, \ldots$
3.   If $X_m \in \mathrm{DIS}(V)$
4.     $Q \leftarrow Q \cup \{X_m\}$
5.   If $|Q| = \lfloor n/k \rfloor$
6.     Request the labels of examples in $Q$
7.     Let $L$ be the corresponding labeled examples
8.     $V \leftarrow \{h \in V : (\mathrm{er}(h; L) - \min_{g \in V} \mathrm{er}(g; L)) \frac{\lfloor n/k \rfloor}{m - m_b} \le \mathcal{E}_{m - m_b}\}$
9.     $b \leftarrow b + 1$ and $Q \leftarrow \emptyset$
10.    $m_b \leftarrow m$
11.    If $b > k$, Return any $\hat{h} \in V$

---

**Theorem 4.2.** *Under the Tsybakov noise condition, letting $\beta = \frac{\alpha}{2 - \alpha}$, and $\bar{\beta} = \sum_{i=0}^{k-1} \beta^i$, for some*

$$\lambda(\epsilon, \delta) = O\left( k \left(\frac{1}{\epsilon}\right)^{\frac{2-\alpha}{\bar{\beta}}} (c_2 \theta(c_2 \epsilon^\alpha))^{1 - \frac{\beta^{k-1}}{\bar{\beta}}} \left( d \log\left(\frac{d}{\epsilon}\right) + \log\left(\frac{kd}{\delta \epsilon}\right) \right)^{\frac{1 + \beta \bar{\beta} - \beta^k}{\bar{\beta}}} \right),$$

*for any $n \ge \lambda(\epsilon, \delta)$, with probability at least $1 - \delta$, running $k$-batch Robust CAL with budget $n$ produces a classifier $\hat{h}$ with $\mathrm{er}(\hat{h}) - \mathrm{er}(h^*) \le \epsilon$.*

*Proof.* Let $M = \lfloor n/k \rfloor$. Define $i_{0M} = 0$ and $V_0 = \mathbb{C}$. Generally, for $b \ge 1$, let $i_{b1}, i_{b2}, \ldots, i_{bM}$ denote the indices $i$ of the first $M$ points $X_i \in \mathrm{DIS}(V_{b-1})$ for which $i > i_{(b-1)M}$, and let $Q_b = \{(X_{i_{b1}}, Y_{i_{b1}}), \ldots, (X_{i_{bM}}, Y_{i_{bM}})\}$ and $V_b = \{h \in V_{b-1} : (\mathrm{er}(h; Q_b) - \min_{g \in V_{b-1}} \mathrm{er}(g; Q_b)) \frac{M}{i_{bM} - i_{(b-1)M}} \le \mathcal{E}_{i_{bM} - i_{(b-1)M}}\}$. These correspond to the set $V$ at the conclusion of batch $b$ in the $k$-batch Robust CAL algorithm.

For $b \in \{1, \ldots, k\}$, (1) (applied under the conditional distribution given $V_{b-1}$, combined with the law of total probability) implies that $\forall m > 0$, letting $Z_{b,m} = \{(X_{i_{(b-1)M}+1}, Y_{i_{(b-1)M}+1}), \ldots, (X_{i_{(b-1)M}+m}, Y_{i_{(b-1)M}+m})\}$, with probability at least $1 - \delta/(4km^2)$, if $h^* \in V_{b-1}$, then $\mathrm{er}(h^*; Z_{b,m}) - \min_{g \in V_{b-1}} \mathrm{er}(g; Z_{b,m}) < \mathcal{E}_m$, and every $h \in V_{b-1}$ with $\mathrm{er}(h; Z_{b,m}) - \min_{g \in V_{b-1}} \mathrm{er}(g; Z_{b,m}) \le \mathcal{E}_m$ has $\mathrm{er}(h) - \mathrm{er}(h^*) < 2\mathcal{E}_m$. By a union bound, this holds for *all* $m \in \mathbb{N}$, with probability at least $1 - \delta/(2k)$. In particular, this means it holds for $m = i_{bM} - i_{(b-1)M}$. But note that for this value of $m$, any $h, g \in V_{b-1}$ have $\mathrm{er}(h; Z_{b,m}) - \mathrm{er}(g; Z_{b,m}) = (\mathrm{er}(h; Q_b) - \mathrm{er}(g; Q_b)) \frac{M}{m}$ (since for every $(x, y) \in Z_{b,m} \setminus Q_b$, either both $h$ and $g$ make a mistake, or neither do). Thus if $h^* \in V_{b-1}$, we have $h^* \in V_b$ as well, and furthermore $\sup_{h \in V_b} \mathrm{er}(h) - \mathrm{er}(h^*) < 2\mathcal{E}_{i_{bM} - i_{(b-1)M}}$. By induction (over $b$) and a union bound, these are satisfied for all $b \in \{1, \ldots, k\}$ with probability at least $1 - \delta/2$. For the remainder of the proof, we suppose this $1 - \delta/2$ probability event occurs.

Next, we focus on lower bounding $i_{bM} - i_{(b-1)M}$, again by induction. As a base case, we clearly have $i_{1M} - i_{0M} \ge M$. Now suppose some $b \in \{2, \ldots, k\}$ has $i_{(b-1)M} - i_{(b-2)M} \ge T_{b-1}$ for some $T_{b-1}$. Then, by the above, we have $\sup_{h \in V_{b-1}} \mathrm{er}(h) - \mathrm{er}(h^*) < 2\mathcal{E}_{T_{b-1}}$. By the Tsybakov noise condition, this implies $V_{b-1} \subseteq \mathrm{B}\left(h^*, c_2 \left(2\mathcal{E}_{T_{b-1}}\right)^\alpha\right)$, so that if $\sup_{h \in V_{b-1}} \mathrm{er}(h) - \mathrm{er}(h^*) > \epsilon$, $P(\mathrm{DIS}(V_{b-1})) \le \theta(c_2 \epsilon^\alpha) c_2 \left(2\mathcal{E}_{T_{b-1}}\right)^\alpha$. Now note that the conditional distribution of $i_{bM} - i_{(b-1)M}$ given $V_{b-1}$ is a negative binomial random variable with parameters $M$ and $1 - P(\mathrm{DIS}(V_{b-1}))$ (that is, a sum of $M$ Geometric$(P(\mathrm{DIS}(V_{b-1})))$ random variables). A Chernoff bound (applied under the conditional distribution given $V_{b-1}$) implies that $P(i_{bM} - i_{(b-1)M} <$

$M/(2P(\text{DIS}(V_{b-1})))|V_{b-1}) < e^{-M/6}$. Thus, for $V_{b-1}$ as above, with probability at least $1 - e^{-M/6}$, $i_{bM} - i_{(b-1)M} \geq \frac{M}{2\theta(c_2\epsilon^\alpha)c_2(2\mathcal{E}_{T_{b-1}})^\alpha}$. Thus, we can define $T_b$ as in the right hand side, which thereby defines a recurrence. By induction, with probability at least $1 - ke^{-M/6} > 1 - \delta/2$,

$$i_{kM} - i_{(k-1)M} \geq M^{\bar{\beta}} \left( \frac{1}{4c_2\theta(c_2\epsilon^\alpha)} \right)^{\bar{\beta}-\beta^{k-1}} \left( \frac{1}{2(d\log(M) + \log(kM/\delta))} \right)^{\beta(\bar{\beta}-\beta^{k-1})}.$$

By a union bound, with probability $1 - \delta$, this occurs simultaneously with the above $\sup_{h \in V_k} \text{er}(h) - \text{er}(h^*) < 2\mathcal{E}_{i_{kM} - i_{(k-1)M}}$ bound. Combining these two results yields

$$\sup_{h \in V_k} \text{er}(h) - \text{er}(h^*) = O\left( \left( \frac{(c_2\theta(c_2\epsilon^\alpha))^{\bar{\beta}-\beta^{k-1}}}{M^{\bar{\beta}}} \right)^{\frac{1}{2-\alpha}} (d\log(M) + \log(kM/\delta))^{\frac{1+\beta(\bar{\beta}-\beta^{k-1})}{2-\alpha}} \right).$$

Setting this to $\epsilon$ and solving for $n$, we find that it suffices to have

$$M \geq c_4 \left( \frac{1}{\epsilon} \right)^{\frac{2-\alpha}{\bar{\beta}}} (c_2\theta(c_2\epsilon^\alpha))^{1-\frac{\beta^{k-1}}{\bar{\beta}}} \left( d\log\left( \frac{d}{\epsilon} \right) + \log\left( \frac{kd}{\delta\epsilon} \right) \right)^{\frac{1+\beta\bar{\beta}-\beta^k}{\bar{\beta}}},$$

for some constant $c_4 \in [1, \infty)$, which then implies the stated result. $\qquad\square$

Note: the threshold $\mathcal{E}_m$ in $k$-batch Robust CAL has a direct dependence on the parameters of the Tsybakov noise condition. We have expressed the algorithm in this way only to simplify the presentation. In practice, such information is not often available. However, we can replace $\hat{\mathcal{E}}_m$ with a data-dependent local Rademacher complexity bound $\hat{\mathcal{E}}_m$, as in [7], which also satisfies (1), and satisfies (with high probability) $\hat{\mathcal{E}}_m \leq c'\mathcal{E}_m$, for some constant $c' \in [1, \infty)$ (see [13]). This modification would therefore provide essentially the same guarantee stated above (up to constant factors), without having any direct dependence on the noise parameters, and the analysis gets only slightly more involved to account for the confidences in the concentration inequalities for these $\hat{\mathcal{E}}_m$ estimators. A similar result can also be obtained for batch-based variants of other noise-robust disagreement-based active learning algorithms from the literature (e.g., a variant of $A^2$ [5] that uses updates based on quantities related to these $\hat{\mathcal{E}}_m$ estimators, in place of the traditional upper-bound/lower-bound construction, would also suffice).

When $k = 1$, Theorem 4.2 matches the best results for passive learning (up to log factors), which are known to be minimax optimal (again, up to log factors). If we let $k$ become large (while still considered as a constant), our result converges to the known results for one-at-a-time active learning with RobustCAL (again, up to log factors) [7, 14]. Although those results are not always minimax optimal, they do represent the state-of-the-art in the general analysis of active learning, and they are really the best we could hope for from basing our algorithm on RobustCAL.

## 5   Buy-in-Bulk Solutions to Cost-Adaptive Active Learning

The above sections discussed scenarios in which we have a fixed number $k$ of batches, and we simply bounded the label complexity achievable within that constraint by considering a variant of CAL that uses $k$ equal-sized batches. In this section, we take a slightly different approach to the problem, by going back to one of the motivations for using batch-based active learning in the first place: namely, sublinear *costs* for answering batches of queries at a time. If the cost of answering $m$ queries at once is sublinear in $m$, then batch-based algorithms arise naturally from the problem of optimizing the total cost required for learning.

Formally, in this section, we suppose we are given a cost function $c : (0, \infty) \rightarrow (0, \infty)$, which is nondecreasing, satisfies $c(\alpha x) \leq \alpha c(x)$ (for $x, \alpha \in [1, \infty)$), and further satisfies the condition that for every $q \in \mathbb{N}$, $\exists q' \in \mathbb{N}$ such that $2c(q) \leq c(q') \leq 4c(q)$, which typically amounts to a kind of smoothness assumption. For instance, $c(q) = \sqrt{q}$ would satisfy these conditions (as would many other smooth increasing concave functions); the latter assumption can be generalized to allow other constants, though we only study this case below for simplicity.

To understand the total cost required for learning in this model, we consider the following cost-adaptive modification of the CAL algorithm.

```
Algorithm: Cost-Adaptive CAL(C)
1. Q ← ∅, R ← DIS(C), V ← C, t ← 0
2. Repeat
3.    q ← 1
4.    Do until P(DIS(V)) ≤ P(R)/2
5.        Let q' > q be minimal such that c(q' − q) ≥ 2c(q)
6.        If c(q' − q) + t > C, Return any ĥ ∈ V
7.        Request the labels of the next q' − q examples in DIS(V)
8.        Update V by removing those classifiers inconsistent with these labels
9.        Let t ← t + c(q' − q)
10.       q ← q'
11.   R ← DIS(V)
```

Note that the total cost expended by this method never exceeds the budget argument $C$. We have the following result on how large of a budget $C$ is sufficient for this method to succeed.

**Theorem 5.1.** *In the realizable case, for some*

$$\lambda(\epsilon,\delta) = O\left(c\Big(\theta(\epsilon)\left(d\log(\theta(\epsilon)) + \log(\log(1/\epsilon)/\delta))\right)\Big)\log(1/\epsilon)\right),$$

*for any $C \geq \lambda(\epsilon,\delta)$, with probability at least $1 - \delta$, Cost-Adaptive CAL(C) returns a classifier $\hat{h}$ with $\mathrm{er}(\hat{h}) \leq \epsilon$.*

*Proof.* Supposing an unlimited budget ($C = \infty$), let us determine how much cost the algorithm incurs prior to having $\sup_{h \in V} \mathrm{er}(h) \leq \epsilon$; this cost would then be a sufficient size for $C$ to guarantee this occurs. First, note that $h^* \in V$ is maintained as an invariant throughout the algorithm. Also, note that if $q$ is ever at least as large as $O(\theta(\epsilon)(d\log(\theta(\epsilon)) + \log(1/\delta')))$, then as in the analysis for CAL [7], we can conclude (via the PAC bound of [9]) that with probability at least $1 - \delta'$,

$$\sup_{h \in V} P(h(X) \neq h^*(X)|X \in R) \leq 1/(2\theta(\epsilon)),$$

so that

$$\sup_{h \in V} \mathrm{er}(h) = \sup_{h \in V} P(h(X) \neq h^*(X)|X \in R)P(R) \leq P(R)/(2\theta(\epsilon)).$$

We know $R = \mathrm{DIS}(V')$ for the set $V'$ which was the value of the variable $V$ at the time this $R$ was obtained. Supposing $\sup_{h \in V'} \mathrm{er}(h) > \epsilon$, we know (by the definition of $\theta(\epsilon)$) that

$$P(R) \leq P\left(\mathrm{DIS}\left(\mathrm{B}\left(h^*, \sup_{h \in V'} \mathrm{er}(h)\right)\right)\right) \leq \theta(\epsilon) \sup_{h \in V'} \mathrm{er}(h).$$

Therefore,

$$\sup_{h \in V} \mathrm{er}(h) \leq \frac{1}{2} \sup_{h \in V'} \mathrm{er}(h).$$

In particular, this implies the condition in Step 4 will be satisfied if this happens while $\sup_{h \in V} \mathrm{er}(h) > \epsilon$. But this condition can be satisfied at most $\lceil \log_2(1/\epsilon) \rceil$ times while $\sup_{h \in V} \mathrm{er}(h) > \epsilon$ (since $\sup_{h \in V} \mathrm{er}(h) \leq P(\mathrm{DIS}(V))$). So with probability at least $1 - \delta'\lceil \log_2(1/\epsilon) \rceil$, as long as $\sup_{h \in V} \mathrm{er}(h) > \epsilon$, we always have $c(q) \leq 4c(O(\theta(\epsilon)(d\log(\theta(\epsilon)) + \log(1/\delta')))) \leq O(c(\theta(\epsilon)(d\log(\theta(\epsilon)) + \log(1/\delta'))))$. Letting $\delta' = \delta/\lceil \log_2(1/\epsilon) \rceil$, this is $1 - \delta$. So for each round of the outer loop while $\sup_{h \in V} \mathrm{er}(h) > \epsilon$, by summing the geometric series of cost values $c(q' - q)$ in the inner loop, we find the total cost incurred is at most $O(c(\theta(\epsilon)(d\log(\theta(\epsilon)) + \log(\log(1/\epsilon)/\delta))))$. Again, there are at most $\lceil \log_2(1/\epsilon) \rceil$ rounds of the outer loop while $\sup_{h \in V} \mathrm{er}(h) > \epsilon$, so that the total cost incurred before we have $\sup_{h \in V} \mathrm{er}(h) \leq \epsilon$ is at most $O(c(\theta(\epsilon)(d\log(\theta(\epsilon)) + \log(\log(1/\epsilon)/\delta)))\log(1/\epsilon))$. □

Comparing this result to the known label complexity of CAL, which is (from [7])

$$O\left(\theta(\epsilon)\left(d\log(\theta(\epsilon)) + \log(\log(1/\epsilon)/\delta)\right)\log(1/\epsilon)\right),$$

we see that the major factor, namely the $O\left(\theta(\epsilon)\left(d\log(\theta(\epsilon)) + \log(\log(1/\epsilon)/\delta)\right)\right)$ factor, is now inside the argument to the cost function $c(\cdot)$. In particular, when this cost function is *sublinear*, we

expect this bound to be significantly smaller than the cost required by the original fully-sequential CAL algorithm, which uses batches of size 1, so that there is a significant advantage to using this batch-mode active learning algorithm.

Again, this result is formulated for the realizable case for simplicity, but can easily be extended to the Tsybakov noise model as in the previous section. In particular, by reasoning quite similar to that above, a cost-adaptive variant of the Robust CAL algorithm of [14] achieves error rate $\mathrm{er}(\hat{h}) - \mathrm{er}(h^*) \le \epsilon$ with probability at least $1 - \delta$ using a total cost

$$O\left(c\Big(\theta(c_2\epsilon^\alpha)c_2^2\epsilon^{2\alpha-2}d\mathrm{polylog}\left(1/(\epsilon\delta)\right)\Big)\log\left(1/\epsilon\right)\right).$$

We omit the technical details for brevity. However, the idea is similar to that above, except that the update to the set $V$ is now as in $k$-batch Robust CAL (with an appropriate modification to the $\delta$-related logarithmic factor in $\mathcal{E}_m$), rather than simply those classifiers making no mistakes. The proof then follows analogous to that of Theorem 5.1, the only major change being that now we bound the number of unlabeled examples processed in the inner loop before $\sup_{h\in V} P(h(X) \ne h^*(X)) \le P(R)/(2\theta)$; letting $V'$ be the previous version space (the one for which $R = \mathrm{DIS}(V')$), we have $P(R) \le \theta c_2(\sup_{h\in V'} \mathrm{er}(h) - \mathrm{er}(h^*))^\alpha$, so that it suffices to have $\sup_{h\in V} P(h(X) \ne h^*(X)) \le (c_2/2)(\sup_{h\in V'} \mathrm{er}(h) - \mathrm{er}(h^*))^\alpha$, and for this it suffices to have $\sup_{h\in V} \mathrm{er}(h) - \mathrm{er}(h^*) \le 2^{-1/\alpha} \sup_{h\in V'} \mathrm{er}(h) - \mathrm{er}(h^*)$; by inverting $\mathcal{E}_m$, we find that it suffices to have a number of samples $\tilde{O}\left((2^{-1/\alpha}\sup_{h\in V'}\mathrm{er}(h) - \mathrm{er}(h^*))^{\alpha-2}d\right)$. Since the number of label requests among $m$ samples in the inner loop is roughly $\tilde{O}(mP(R)) \le \tilde{O}(m\theta c_2(\sup_{h\in V'}\mathrm{er}(h) - \mathrm{er}(h^*))^\alpha)$, the batch size needed to make $\sup_{h\in V} P(h(X) \ne h^*(X)) \le P(R)/(2\theta)$ is at most $\tilde{O}\left(\theta c_2 2^{2/\alpha}(\sup_{h\in V'}\mathrm{er}(h) - \mathrm{er}(h^*))^{2\alpha-2}d\right)$. When $\sup_{h\in V'}\mathrm{er}(h) - \mathrm{er}(h^*) > \epsilon$, this is $\tilde{O}\left(\theta c_2 2^{2/\alpha}\epsilon^{2\alpha-2}d\right)$. If $\sup_{h\in V} P(h(X) \ne h^*(X)) \le P(R)/(2\theta)$ is ever satisfied, then by the same reasoning as above, the update condition in Step 4 would be satisfied. Again, this update can be satisfied at most $\log(1/\epsilon)$ times before achieving $\sup_{h\in V}\mathrm{er}(h) - \mathrm{er}(h^*) \le \epsilon$.

# 6 Conclusions

We have seen that the analysis of active learning can be adapted to the setting in which labels are requested in *batches*. We studied this in two related models of learning. In the first case, we supposed the number $k$ of batches is specified, and we analyzed the number of label requests used by an algorithm that requested labels in $k$ equal-sized batches. As a function of $k$, this label complexity became closer to that of the analogous results for fully-sequential active learning for larger values of $k$, and closer to the label complexity of passive learning for smaller values of $k$, as one would expect. Our second model was based on a notion of the *cost* to request the labels of a batch of a given size. We studied an active learning algorithm designed for this setting, and found that the total cost used by this algorithm may often be significantly smaller than that used by the analogous fully-sequential active learning methods, particularly when the cost function is *sublinear*.

There are many active learning algorithms in the literature that can be described (or analyzed) in terms of batches of label requests. For instance, this is the case for the margin-based active learning strategy explored by [15]. Here we have only studied variants of CAL (and its noise-robust generalization). However, one could also apply this style of analysis to other methods, to investigate analogous questions of how the label complexities of such methods degrade as the batch sizes increase, or how such methods might be modified to account for a sublinear cost function, and what results one might obtain on the total cost of learning with these modified methods. This could potentially be a fruitful future direction for the study of batch mode active learning.

The tradeoff between the total number of queries and the number of rounds examined in this paper is natural to study. Similar tradeoffs have been studied in other contexts. In any two-party communication task, there are three measures of complexity that are typically used: communication complexity (the total number of bits exchanged), round complexity (the number of rounds of communication), and time complexity. The classic work [16] considered the problem of the tradeoffs between communication complexity and rounds of communication. [17] studies the tradeoffs among all three of communication complexity, round complexity, and time complexity. Interested readers may wish to go beyond the present and to study the tradeoffs among all the three measures of complexity for batch mode active learning.

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
