[Reviews · NeurIPS 2013]

Submitted by Assigned_Reviewer_3

The author(s) are essentially extending [Hanneke, Rates of Convergence in Active Learning; 2011] to batch-mode active learning. Specifically, there are two results of note: (1) that the label complexity of batch (buy-in-bulk) active learning interpolates between passive learning (seeing all of the labels at once) and standard sequential active learning (one queries example at time) -- both for the realizable case and with Tsybakov noise and (2) if the cost of buy-in-bulk active learning is sublinear in the size of the batch, the total cost may be smaller than that of sequential active learning at a similar cost (which obviously makes intuitive sense).

I will address the {quality, clarity, originality, significance} components separately below.

Quality: After significant effort on my part, I am fairly certain that the proofs are correct. The work is well-contextualized, motivated, and answers meaningful theoretical questions. My only comment on this front might be to include other "batch mode" active learning works just for context (as this is one place where you have more space) and practitioners may be interested in comparing this to more heuristic methods (since this is what they will actually implement).

Clarity: Once I was able to internalize the notation and do a bit of background reading, the proofs were actually fairly straightforward (and very clear).

Originality: Once you read [Hanneke, 2011], it is fairly straightforward to get through this paper, so this represents more of a "finding a good questions to ask" paper as opposed to introducing new methods of analysis. However, I haven't seen a good theoretical presentation of batch-mode active learning, so this will almost certainly further the understanding of the active learning community.

Significance: As previously stated, this is more of a "extending theoretical results to a related setting" as opposed to developing new methods for understanding active learning (but this is the difference between a good paper and a "best" paper. I don't think this will have tremendous significance, but is an important advancement in the literature that people in this community will find useful and interesting.

Minor comment
-----
- pg 1 - I assume "match maximum" is supposed to be "batch maximum", no? Otherwise, I am missing something
Summary: This is the first good theoretical presentation of batch mode ("buy-in-bulk") active learning with respect to label complexity. It analyzes buy-in-bulk active learning under realizable cases, noisy label cases, and settings of various label costs. It is well-motivated, sound, and interesting.

Submitted by Assigned_Reviewer_5

Summary of Paper

The paper studies active learning problems in which labels can be acquired in batches, with sublinear labeling costs for labels within a batch. Specifically, the paper studies finite-sample risk bounds for settings in which the model space is assumed to contain a model with risk zero (implying well-specified models and zero noise) and a relaxed version that allows for Tsybakov noise.


Evaluation

The presented setting in which we assume sublinear labeling costs for labels within a batch is interesting and relevant for active learning. This setting has been studied in the literature e.g. by formulating label selection as an optimization problem (Chakraborty et al., 2010). In contrast, the current paper tackles the problem by modifying the PAC-based active learning algorithm CAL (Cohn et al., 1994), and is focused on proving finite-sample bounds in this setting. I am not aware of any other work that derives finite-sample bounds for batch-based active learning, thus this appears to be a novel type of analysis in an established setting.

The first part of the paper analyzes simple extensions of CAL that obtain labels in k equal-sized batches, showing that the resulting label complexity lies in between passive learning and the original CAL algorithm. I feel that this part of the analysis is not very much related to the main motivation of the paper of having a cost function on batches of labels (with sublinear costs). The main contribution of the paper is to adapt the CAL algorithm to such costs functions on batches. For this setting, a detailed proof is provided for the realizable case, and a proof sketch is provided for the case including Tsybakov noise. The analysis here is interesting but relies on certain assumptions about the relationship between D_{X,Y} and the hypothesis space (in the form of the Tsybakov noise model). To understand how well this matches real-world learning tasks, I think a careful empirical evaluation would be very helpful, which is unfortunately missing. For example, the proposed algorithm could be compared against standard active learning and some of the batch-optimized active learners recently proposed (eg, Chakraborty et al, 2010).

Overall, I have the impression that the paper to some degree falls short of its own goals: the setting of buying labels with batch-level costs functions is well motivated with realistic application domains, but the resulting algorithms are never evaluated empirically on such domains.

The paper is mostly well-written and the general ideas presented are quite easy to follow. The theoretical analysis is spelled out in sufficient detail. Proofs of theorems could be moved to an (online) appendix if there is need to free up some space (eg, for an empirical study).
Summary: A paper that derives finite sample bounds for an active learning setting in which labels can be acquired in batches with sublinear costs. Unfortunately, the theoretical analysis is not backed up by empirical results.

Submitted by Assigned_Reviewer_6

This paper introduces a new aspect to the theory of active learning: querying-in-bulk. The setting is well motivated by practical applications and clearly defined. The authors then adapt a basic active learner (CAL) to this new setting in three variations: Realizable case, Tsybacov noise and under a specified cost function for batches of queries. The authors provide performance guarantee with query bounds in terms of the disagreement coefficient for all three cases. In the first two cases their bound interpolates between passive learning and standard active learning for the respective learning setting.

The results are solid and well presented and the setting is new. However, being an adaptation of a existing algorithm, presented with disagreement-coefficient depended bounds, it does not provide significant new insights for our understanding of active learning. So the progress seems rather incremental.
Summary: This is a solid (but not overly exciting) work on the theory of active learning. It introduces a new setting of querying-in-bulk, adapts a well-known active learner to this setting and provides performance guarantees.

Submitted by Assigned_Reviewer_8

*Summary*

This paper discusses (disagreement-based) batch-mode active learning, and puts forward a theoretical framework bounding the sample complexity and the total cost of buy-in-bulk active learning (i.e., batch-mode active learning with sublinear cost functions w.r.t. the size of a batch). In particular, they study batch-mode active learning from two aspects: (1) Suppose the batch sizes are equal. Fix the number of rounds examined, how many labels one has to request in total to achieve some certain error rate; (2) given that the cost to obtain a batch of labels is sublinear in the size of the batch ( as referred to as ``buy-in-bulk discount''), how is the total cost of the proposed batch algorithms compared with that of fully-sequential active learning methods.

For the first aspect, the authors propose batch-based variants of the well-known CAL algorithm for sequential active learning, and provide upper bounds on label complexity of k-batch active learning, for both the realizable case and the non-realizable case (with Tsybakov noise). For the second aspect, they provide a cost-adaptive modification of the CAL algorithm, and find that the total cost by this algorithm may often be significantly smaller than that of the analogous methods in the fully sequential setting.

*Quality*

This paper nicely extends the previous work (Hanneke ’10) to the batch-mode setting. The first part (section 3, 4) derives upper bounds (depending the size of a batch) of label complexity of the proposed batch algorithms. It is technically sound, and thoroughly explained. The second part (section 5), however, is less clear. From line 376 to 380, the authors directly compare two upper bounds that are obtained separately (potentially with different low-order terms hidden in the big O notation), and conclude that buy-in-bulk active learning are usually more cost-efficient - which seems not fair to me.

Besides, there is no empirical evaluation of the proposed algorithm. Given the various motivating applications mentioned before, it would be interesting to see how the Cost-Adaptive CAL performs empirically with different cost models.

*Clarity*

The second result (Section 5) for buy-in-bulk active learning, as appears in the paper, is striking, yet unconvincing. According to Theorem 5.1, if the cost function is linear in batch sizes, then the proposed batch-mode algorithm (Cost-Adaptive CAL) will exhibit the same form of upper bound as that of the fully-sequential CAL algorithm. Intuitively, under such (unit) cost setting, the cost of Cost-Adaptive CAL should be higher than fully sequential CAL. However, according to the two upper bounded provided in line 343 and 367, one can not tell the difference. Further analysis on the difference (hidden constant or low-order terms) between the cost bounds would be helpful.

Originality and Significance
Active learning problems under the batch-mode setting are quite natural and interesting, but so far only a few theoretical work have been done (e.g., Chen & Krause, ICML’13 formulate it as an adaptive optimization problem). This paper explores the theoretical aspect of the mellow schemes for batch-mode active learning. Although the proof technique relies heavily on the previous work on (sequential) active learning, it is one of the few attempts made along this line of research.

Editorial issue(s):
1. line 171, second term in the max bracket: $er(h)$ is not a subscript.
2. the paper exceeds the submission page limit by 3 lines.

Update: I've considered the authors' rebuttal, and decide not to change my score. Specifically, I’m still not convinced that the cost of the batch algorithm is only constant factor away from the sequential CAL (according to the rebuttal) -- there might be some lower order terms in the bound of the batch algorithm, which is important when comparing with the sequential bound. This is one of the main reasons that I would like to see some empirical evidence of the statement.
Summary: Overall, I find the paper theoretically interesting. It generalize the result of [Hanneke ’10] to the batch-mode setting. The results seem correct. However, the theoretical analysis for buy-in-bulk solution (section 5), which is more closely-related to the theme of the paper, is less clear (as explained previously). Besides, no empirical study (especially for the setting in section 5) is conducted, which I think is important given the practical importance of the problem setting.
Author Feedback

Author rebuttal: We thank all of the reviewers for these helpful
comments. Below are our responses to a few specific
points raised in the reviews.

Assigned_Reviewer_5 mentions that the results for
sublinear cost functions are only given for the
realizable case. However, we note that the paper
does include such a result under Tsybakov noise on
page 8. We chose to abbreviate the proof due to the
space limitation, but the ideas are sketched there.
Once accepted, we will provide the full details of
the proof in an extended version of the paper for
the arXiv.

Assigned_Reviewer_8 wonders about the comparison
between upper bounds for the full-sequential vs
cost-adaptive variants of CAL. We acknowledge that
an improvement in upper bound does not always
reflect an improvement in the actual cost complexity;
however, in this case, it is clear that such
an improvement actually does occur in certain cases,
especially when the cost is much smaller than linear.
For instance, if the cost is c(x)=x^{1/2}, we reduce
by a factor proportional to (theta*d)^{1/2}. It is
true that the constant factors can be slightly larger
in the cost-adaptive bound (say, by a factor of 4),
but for large d or theta, we will still have a net
reduction in total cost.

As for the question of empirical comparisons to other
batch-based methods, this work is strictly
theoretical, and we do not plan on including an
empirical evaluation.